# New Insights into the Multivariate Analysis of SER Spectra Collected on Blood Samples for Prostate Cancer Detection: Towards a Better Understanding of the Role Played by Different Biomolecules on Cancer Screening: A Preliminary Study

**DOI:** 10.3390/cancers14133227

**Published:** 2022-06-30

**Authors:** Vlad Cristian Munteanu, Raluca Andrada Munteanu, Diana Gulei, Radu Mărginean, Vlad Horia Schițcu, Anca Onaciu, Valentin Toma, Gabriela Fabiola Știufiuc, Ioan Coman, Rareș Ionuț Știufiuc

**Affiliations:** 1Department of Urology, The Oncology Institute “Prof Dr. Ion Chiricuta”, 400015 Cluj-Napoca, Romania; vladcristian.munteanu@gmail.com (V.C.M.); schitcu@yahoo.com (V.H.S.); 2Department of Urology, “Iuliu Hatieganu” University of Medicine and Pharmacy, 400012 Cluj-Napoca, Romania; jcoman@yahoo.com; 3“Iuliu Hatieganu” University of Medicine and Pharmacy, 400012 Cluj-Napoca, Romania; muresan.raluca.andrada@gmail.com (R.A.M.); anca.onaciu@umfcluj.ro (A.O.); 4MedFuture—Research Center for Advanced Medicine, “Iuliu Hatieganu” University of Medicine and Pharmacy, 400337 Cluj-Napoca, Romania; diana.c.gulei@gmail.com (D.G.); margi.radu@outlook.com (R.M.); valentin.toma@umfcluj.ro (V.T.); 5Faculty of Physics, “Babes Bolyai” University, 400084 Cluj-Napoca, Romania; gabriela.stiufiuc@ubbcluj.ro; 6Department of Urology, Clinical Municipal Hospital, 400139 Cluj-Napoca, Romania; 7Department of Pharmaceutical Physics-Biophysics, “Iuliu Hațieganu” University of Medicine and Pharmacy, 400349 Cluj-Napoca, Romania

**Keywords:** prostate cancer, Raman, SERS, multivariate analysis

## Abstract

**Simple Summary:**

In recent years, research on biofluids using Raman and SERS has expanded dramatically, indicating the enormous promise of this technology as a high-throughput tool for identifying cancer and other disorders. In the investigations thus far, researchers have concentrated on a specific illness or condition, but the techniques employed to acquire experimental spectra prevent direct comparison of the data. This necessitates comparative research of a variety of diseases and an increase in scientific cooperation to standardize experimental conditions. In our study, positive results were reached by applying a combined SERS multivariate analysis (MVA) to the urgent problem of prostate cancer diagnosis that was directly linked to real-world settings in healthcare. Moreover, in comparison to the prostate-specific antigen (PSA) test, which has a high sensitivity but limited specificity, our combined SERS-MVA method has greater specificity, which may assist in preventing the overtreatment of patients.

**Abstract:**

It is possible to obtain diagnostically relevant data on the changes in biochemical elements brought on by cancer via the use of multivariate analysis of vibrational spectra recorded on biological fluids. Prostate cancer and control groups included in this research generated almost similar SERS spectra, which means that the values of peak intensities present in SERS spectra can only give unspecific and limited information for distinguishing between the two groups. Our diagnostic algorithm for prostate cancer (PCa) differentiation was built using principal component analysis and linear discriminant analysis (PCA-LDA) analysis of spectral data, which has been widely used in spectral data management in many studies and has shown promising results so far. In order to fully utilize the entire SERS spectrum and automatically determine the most meaningful spectral features that can be used to differentiate PCa from healthy patients, we perform a multivariate analysis on both the entire and specific spectral intervals. Using the PCA-LDA model, the prostate cancer and control groups are clearly distinguished in our investigation. The separability of the following two data sets is also evaluated using two alternative discrimination techniques: principal least squares discriminant analysis (PLS-DA) and principal component analysis—support vector machine (PCA-SVM).

## 1. Introduction

Prostate cancer is a major public health problem. It represents the second most diagnosed neoplasm and occupies sixth place in terms of mortality. In 2018, there were approximately 1.3 million cases and 359,000 deaths worldwide due to prostate cancer (PCa). In Europe, the estimated incidence of PCa in the same year was 449,800 cases and 107,300 deaths. This trend is stationary in many countries and is in a slow decline in high-income countries [1].

Before the discovery of prostatic specific antigen (PSA) in the 1970s and screening studies in the late 1980s, there was no way of screening for prostate cancers. Most of the patients first presented with metastatic disease, because nonmetastatic tumors are asymptomatic. Once PSA was discovered and used on a global scale, PCa became curable [2]. As such, urologists introduced new PSA-based screening procedures for PCa detection and soon started overdiagnosing and overtreating not only aggressive cases but cases that later proved to be indolent cancers. Unfortunately, PSA is organ-specific and not disease-specific, having high sensitivity but low specificity. PSA-based screening tests identify a lot of indolent cancers and have minimal impact on identifying aggressive tumors. To this day, PSA represents the cornerstone of prostate cancer diagnosis [3], and the ultimate goal remains to identify and treat only aggressive cancers [4,5].

The classical PCa detection scenario, based on PSA and prostate biopsy, has a detection rate of 20–40% accuracy [6], which is quite low compared to the incidence of this disease. In recent years, a lot of new alternative diagnostic modalities aroused such as blood and urine tests, and imaging modalities. Some of them even proved to be superior to PSA in detecting significant PCa cases [7]. Still, the challenge is to find a reliable, affordable, and accurate biomarker [8].

On the other hand, one of the most promising tools in the arsenal of developing new strategies for PCa diagnosis is Raman and its counterpart, surface-enhanced Raman spectroscopy (SERS). Raman spectroscopy has the capacity to provide specific molecular data (molecular fingerprint) that could have a major impact in the medical field, such as assisting new biomarker identification regarding cancer development [9]. These optical techniques are based on the inelastic scattering of the photons after the monochromatic laser beam interacts with specific molecules present in the biological sample. The difference between the energy of the photons before and after interacting with the sample, measured in wavenumbers, represents the Raman shift. These shifts, taken together, form the Raman spectrum, with each peak being assigned to a specific vibrational mode encountered in the sample [10]. Although Raman spectroscopy is able to detect a considerable number of biological molecules and offers support in the medical diagnosis area, its applicability can be limited by analyte concentration, which affects the intensity of the signal. Moreover, depending on the protocol strategy, the distribution of the molecules will not be homogenous, and the spectral bands will be preponderately assigned to proteins and other high molecular weight biomolecules present in the sample [11].

In the case of SERS, the procedure implies the use of metallic plasmonic substrates whose role is to enhance the Raman signal of the molecules present in the very close vicinity (<10 nm) of the plasmonic nanoparticles that compose the substrates. Depending on the adsorption geometry of the sample molecules onto these surfaces, their bands’ intensity varies, which slightly complicates spectra interpretation. Several SERS-based cancer studies performed on blood samples derivatives reported an accuracy of over 90% in differentiating between PCa groups and controls [12,13,14]. Silver and/or gold nanoparticles are widely used as plasmonic substrates for such investigations. By carefully engineering substrates’ composition and morphology, it was shown that SERS has the capacity to identify nanoscale molecular interactions responsible for chiral discrimination [15,16,17]. The use of low concentrations analytes is another major advantage of SERS analysis. However, similar to in most of the cases, these advantages come with a cost, and in the case of SERS performed on biological samples, the most important drawback is the lack of signal reproducibility. Very recently, our research group developed a new type of plasmonic solid substrate based on tangential flow filtered (TFF) silver nanoparticles capable of generating reproducible spectra that have been further analyzed by means of multivariate analysis (MVA) in order to develop an early-stage diagnostic tool for breast cancer [18].

In the last years, our research group has demonstrated that a combined SERS-MVA analysis can be successfully applied for the diagnosis of different types of solid tumors [19], including prostate cancer [11], using serum samples collected from cancer patients. Moreover, such implementations have been extensively involved in various statistical algorithms used to differentiate between normal and cancerous tissue from biopsies [20,21]. It has been shown that a SERS analysis on serum samples was able to discriminate between prostate cancer and benign prostatic hyperplasia (BPH) in an attempt to decrease the number of unnecessary biopsies [12]. In these cases, the SERS substrate was used in a colloidal formulation.

However, many difficulties can be encountered in the case of biofluid analysis due to their complex molecular composition. Very recently, Fornasaro et al., 2021, have shown that ergothioneine, which is a dietary amino acid present in different biological samples, has a great impact on the SERS spectra collected on various biofluids (e.g., erythrocytes lysates, serum, gingival crevicular fluid, seminal plasma, cerebrospinal fluid). This phenomenon may occur due to its high affinity for the plasmonic substrates, highlighting once more the major role played by the nanoscale interactions of the biomolecules with the plasmonic nanostructure in SERS analysis [22].

To overcome this drawback and to try to understand the influence of different molecular species on cancer discrimination using the here-proposed SERS-MVA analysis, in this study, we have employed a twofold strategy. Firstly, we have recorded, using our solid plasmonic substrates, very reproducible SERS spectra on serum and plasma samples collected from healthy (Controls) and prostate cancer donors (Patients) that were further compared and analyzed by means of MVA. Secondly, complete MVA studies have been performed not only on the entire spectra but also on specific spectral regions where the most intense vibrational bands have been assigned to proteins and/or other biomolecules in order to understand if these vibrational bands can be used for proper discrimination between cancer and control samples. In the end, the separability of the two data sets was evaluated using the following two alternative discrimination techniques: principal least squares discriminant analysis (PLS-DA) and principal component analysis—support vector machine (PCA-SVM).

To the best of our knowledge, such a comprehensive SERS-MVA analysis of vibrational spectra collected on plasma and serum for cancer discrimination has not been reported so far in the scientific literature.

## 2. Materials and Methods

### 2.1. Sample Collection

Between July 2018 and March 2020 we collected blood and prostate tissue samples from 103 patients treated in the Institute of Oncology “Prof. Dr. Ion Chiricuta” in Cluj-Napoca, Romania, in conformity with the ethical accordance 119/20 March 2020 from the University of Medicine and Pharmacy “Iuliu Hatieganu” Cluj-Napoca. All patients were previously diagnosed with prostate cancer through prostate biopsy. We excluded patients with other known diseases or those who had previous prostate cancer treatment (radiotherapy or androgen deprivation therapy). Regarding the PCa patient’s cohort, the average age was 61 (min 52, maximum 68). In the case of the healthy donors’ cohort, were selected individuals who were referred by the general practitioner to perform routine urological check-ups with an average age of over 50 years old.

Collected blood samples were immediately processed. For the processing of plasma, blood samples were immediately centrifuged for 10 min at 4000 rpm and the resulting supernatant (plasma) was transferred to a new tube that was stored at −80 °C until further processing. For serum, the blood collection tubes were left at room temperature for 30 min and then centrifuged for 10 min at 4000 rpm. The supernatant was transferred to a new tube and stored at −80 °C until further processing. All tubes were anonymously annotated based on patient’s codes and additional variables.

### 2.2. Synthesis of Silver Nanoparticles

The silver nanoparticles were synthesized using the protocol developed by Leopold and Lendl, 2003 [23]. All the solutions were prepared using ultrapure water (18.2 MΩ × cm, ELGA Labwater from PURELAB Chorus, Buckinghamshire, UK). Briefly, 5 mL of 30 mM NH_2_OH·HCl solution was mixed with 5 mL of 63.5 mM NaOH and 80 mL ultrapure water under vigorous stirring conditions (400 rpm). Then 10 mL of 10 mM AgNO_3_ solution was carefully incorporated, under continuous stirring for 10 min until it was observed a brown to yellowish coloration. The resulting silver colloid was subjected to tangential flow filtration (TFF, Pall Corporation, New York, NY, USA) and physical characterization for further plasmonic substrate assembly.

### 2.3. SERS Substrates Preparation

Solid plasmonic SERS substrate preparation was performed according to a procedure described by Stiufiuc et al., 2020 [18]. This included several cleaning steps of CaF_2_ Raman grade glass (Crystran, Poole, UK) using acetone, ethanol, and ultrapure water. After 15 min, the port-probe was heated at 40 °C using a plate heater and 1 µL of concentrated silver colloids was added to this site and let dry for 2 min. The obtained solid substrates were ready to use for SERS analysis after cooling down at room temperature.

### 2.4. SERS Measurements

For SERS measurements, 1 µL of serum, respectively, 1 µL plasma, were poured on the top of plasmonic substrates and were left to dry for 30 min at room temperature before acquiring the SERS signal. Both spectra types were recorded at maximum 50 µm distance from the sample ring edges. The analysis was performed using the Renishaw™ inVia Reflex Raman (Renishaw plc, Gloucestershire, UK) confocal multilaser spectrometer at a resolution of 2 cm^−1^. The spectrograph was equipped with a 600 lines/mm grating and a charge-coupled device camera (CCD). An internal silicon reference was used for calibration. The 50× (N.A = 0.75) objective lens was used to record the spectra. A 785 nm diode laser was used for excitation. In the case of SERS, the acquisition time was set at 20 s (exposure time 5 s and 4 accumulations) while the laser power to the surface of the sample was 2 mW. Baseline correction was applied to all SERS spectra in order to eliminate the fluorescence background. The baseline correction was performed by using the Wire 4.2 software provided by Renishaw (Gloucestershire, UK) and final data processing was performed with aid of OriginPro 2019 software platform. The final spectrum represents the average of 20 spectral acquisitions.

### 2.5. Data Analysis

To inspect whether there is a separation between patient and control sample sets, we use a multivariate approach that is suitable for comparing high-dimensional objects such as spectral data. Thus, we apply principal component analysis—linear discriminant analysis (PCA-LDA) to the spectra, a method that combines dimensionality reduction with multivariate classification.

As a preprocessing step for the multivariate analysis, we align the spectra by sampling at equal 1 cm^−1^ intervals and normalizing them using the standard normal variate method, where each spectrum’s intensities are scaled and offset such that they have zero mean and unit standard deviation.

Due to the curse of dimensionality, high-dimensional objects cannot be reliably compared in small samples. For this reason, we show that most of the information contained within our data is contained in a small number of dimensions—the principal components obtained via principal component analysis (PCA). Thus, by using PCA, we project the data onto a low-dimensional space by filtering out the noisy dimensions. This allows us to proceed with linear discriminant analysis (LDA), a method that finds a plane that separates data points belonging to different classes by optimizing for the maximum ratio of between-class and within-class variances.

The resulting LDA plane separates the projected spectra into two classes—patient and control. To assess the quality of this separation, we employ a leave-one-out cross-validation (LOOCV) scheme to efficiently use our relatively small dataset.

For completeness, we also evaluate the separability of the following two spectra sets (in LOOCV fashion) using alternative discrimination techniques: principal least squares discriminant analysis (PLS-DA), and principal component analysis—support vector machine (PCA-SVM).

We also perform a univariate analysis, where we test the separability hypothesis at each sampled wave number using a t-test. To account for multiple testing, we also apply a Benjamini-Hochberg correction with the false discovery rate set at 5% [24].

## 3. Results

### 3.1. Subject Data and Pathological Classification

One hundred and three patients were screened for enrolment in the study. After applying the exclusion criteria based on previous treatments and additional pathological status, as well as sample technical eligibility, 29 PCa patients were included. The clinical data for the patients are summarized in Table 1. The control cohort was formed of 14 samples. For all donors, both serum and plasma samples were analyzed.

### 3.2. SERS Analysis of Plasma and Serum Samples

Plasma and serum SERS spectra were recorded at a maximum 50 µm distance from the analyte ring edges and normalized to the integrated area under the curve in the 350–2200 cm^−1^ spectral interval. Figure 1 shows the average SERS spectra recorded on blood plasma samples collected from healthy and PCa patients. One can notice that the spectra are dominated by the following vibrational peaks: 390, 498, 596, 642, 728, 815, 893, 1010, 1075, 1136, 1209, 1256, 1336, 1369, 1406, 1447, 1508, 1577, 1617, and 1662 cm^−1^. From these, 1256, 1336, 1506, 1617, and 1662 cm^−1^ bands are more intense in the case of the healthy group compared to the PCa group. Three prominent peaks that display the strongest SERS signal among both groups are located at 642, 1136, and 1662 cm^−1^.

The SERS spectra of the serum samples, recorded using exactly the same experimental conditions and substrates, are presented in Figure 2. It can be noticed that the same three peaks dominate the spectra, as in the case of plasma samples. A distinct vibrational band located at 1099 cm^−1^ can be remarked only in the case of serum spectra collected from PCa patients. Moreover, slight differences can be observed regarding the intensity of several bands, which are more prominent for the healthy group (728, 1334, 1447, 1506, 1580, and 1662 cm^−1^) than for the patients’ group.

When visually comparing the plasmatic and serum SERS spectra, one can observe that serum samples offer a better separation between control and PCa groups.

### 3.3. Data Analysis

The PCA-LDA analysis uses two principal components for projecting the spectra. The number of components was chosen for the robustness to noise and amount of information contained within (explained variance). For plasma samples, the two components explain 76% of the variance (99% is explained by 14 components), while for serum samples, the two components explain 78% of the variance (99% is explained by 13 components).

The spectra projected onto the two principal components (PCs) are used as input for the discrimination step of the PCA-LDA analysis. For the plasma set of samples, our dataset consists of 14 controls and 27 patients, while for the serum samples it consists of 14 controls and 29 patients. The results of the PCA-LDA analysis, as evaluated using the LOOCV strategy, are shown in the following table (Table 2).

Additionally, we also performed a PCA-LDA analysis under the same set-up on a restricted band of wavenumbers, between 1200 cm^−1^ and 1700 cm^−1^. Table 3 shows the obtained results.

We ran experiments with more principal components as well, and adding more PCs generally improves the obtained results. Nevertheless, given the size of our dataset, we decided to use a small number of PCs to avoid the potential overfitting of complex multivariate models to our data. Specifically, with two PCs, we see negligible differences in the classification performance obtained via LOOCV and the training folds (Figure 3). With larger numbers of PCs, this difference is more pronounced, suggesting that the more complex PCA models do not generalize as well from our limited data set.

Appendix A shows the accuracy obtained with different models (PCA-LDA and PLSDA) for both train and test samples. We used accuracy for ease of measurement and clarity—the same relative train/test differences can be observed across other metrics as well.

Finally, to validate further the observed separations, we also classify the spectra using principal least squares discriminant analysis (PLSDA) with two intermediate dimensions, similar to our PCA-LDA setup. The following table (Table 4) shows the obtained results.

The use of the PLSDA method allowed us to compute the importance of wavenumbers in the classification decision by using the variable importance in projection (VIP) score, which measures the relative contribution of each variable (wavenumber) in the classification decision. A VIP score greater than 1.0 is conventionally considered to be the threshold for selecting important variables. In the following figures (Figure 3 and Figure 4), we show the bands of important variables as instructed by the VIP score, as well as the bands of wavenumbers where the univariate difference in mean intensity between the patient and control sets is deemed significant by a Benjamini-Hochberg (BH)-corrected t-test using a false discovery rate (FDR) set at 5%.

We also mark the significant peaks identified in these mean spectra. The identified peaks can differ slightly between the class-wise and grouped charts since the control and patient mean spectra can have peaks that do not perfectly overlap (in the charts only one of the peaks is shown for figure legibility), and these will determine a different peak in the grouped spectrum.

For the plasma samples, we see significant peaks around the 390, 1012, 1210, 1260, 1622, and 1666 cm^−1^ spectral regions (Figure 4).

For the serum samples, we see significant peaks around the 390, 499, 643, 729, 815, 892, 1012, 1100, 1137, 1210, 1331, 1368, 1412, 1511, 1582, and 1660 cm^−1^, and in the range above 1700 cm^−1^ spectral bands (Figure 5).

We observe that for both plasma and serum samples, there is a significant overlap between the wavenumbers considered important in the univariate and multivariate analysis.

The classification results of PCA-SVM using 2 PCs and a linear kernel for the SVM, SVM using a linear kernel and no dimensionality reduction, and LDA with no dimensionality reduction are presented in Appendix A. Appendix A show the mean spectrum and the first two principal components of the plasma and serum samples.

## 4. Discussion

Liquid biopsies, including plasma, serum, or urine, offer a valuable platform in determining new biomarkers for prostate cancer diagnosis, leading to time efficiency and enlarging the treatment options, therefore improving the quality of life for such patients.

Cell-free nucleic acids (cfNA), which may be detected in the blood plasma, have been the subject of most liquid biopsy investigations aimed at identifying biomarkers that are predictive, diagnostic, and/or prognostic in cancer [25]. Other elements, such as circulating proteins, analytes, and exosomes, have received less research attention, however. Cancer patients’ circulating DNA, tumor cells, and exosomes may all be detected using liquid biopsy methods that have yet to be used in the clinical setting [26,27,28].

This study’s aim was to explore the outstanding properties of univariate and multivariate analysis performed on plasmatic and serum SERS spectra in discriminating between PCa patients and healthy donors. We also wanted to investigate the role of the most important vibrational bands, assigned to different biomolecules present in blood samples, in the discrimination process.

SERS is an ultrasensitive technique that can achieve a diagnostic value by enabling the spectral analysis of biological samples and molecules. Most of the SERS analyses on blood samples are performed on colloidal nanoparticles. The affinity of the molecules toward the plasmonic substrate plays a crucial role in the recording process of SERS spectra. Very recently, it has been shown that there is a strong possibility that much of the SERS spectra collected on blood samples reported so far in the literature are dominated by a dietary amino acid (ergothioneine) that has a great affinity for the plasmonic substrates used in SERS experiments [22]. In order to reduce the possibility of the occurrence of such experimental artifacts, all the spectra included in this study have been recorded on solid plasmonic substrates prepared using a procedure developed in our laboratory that proved their capacity to generate specific and reproducible SERS spectra of blood plasma and serum based on their ability to act as a “spectroscopic filter” [18].

The multivariate analysis of these spectral data offers the advantage of determining more accurately and realistically the factors that influence the variability between the two groups of samples. In addition, the univariate analysis represents a strong descriptive method that can clearly elucidate the differences between the two types of samples investigated in this study. Moreover, such algorithms are still needed to be implemented in the diagnosis steps for a better correlation with molecular modifications associated with cancer development and progression.

Our multivariate analysis, performed on the entire spectral window (350–2200 cm^−1^), supports the idea that the use of serum samples instead of plasma ones can improve the discrimination process between PCa patients and healthy donors. The results obtained on serum samples offered a better accuracy (97.7% vs. 87.8%), precision (100% vs. 86.7%), sensitivity (96.6% vs. 96.3%), and specificity (100% vs. 71.4%) as compared to those obtained on plasma samples.

We have performed a univariate analysis, where we test the separability hypothesis at each sampled wavenumber using a *t*-test. To account for multiple testing, we also applied a BH correction with the FDR set at 5%. In other words, we have tested the hypothesis that the mean intensity is significantly different between the two groups (control and patients) at each individual wave number. The univariate nature of the analysis is due to the lack of interaction between the different variables (wavenumbers) in the analysis. We have reported our results as charts of the mean spectrum (both class-wise and grouped), where we have emphasized the regions of wave numbers that were identified as being significant with respect to the BH-corrected t-test.

Plasma sample spectral data analysis detected 6 major regions in the SERS spectra corresponding to 350–400, 740–870, 990–1030, 1220–1300, 1390–1410, and 1750–1760 cm^−1^ bands, achieving an AUROC (area under the receiver operating characteristic curve) value of 0.9 or higher. In addition, we also identified three other isolated spectral windows (centered at 970, 1180, and 1710 cm^−1^) that could play a role in discrimination. On the other hand, our univariate analysis identified the presence of the following 6 major peaks relevant for SERS-based discrimination between PCa and controls: 390, 1012, 1210, 1260, 1622, and 1666 cm^−1^. They can be assigned as follows: ~390—uric acid [29], ~1012—breathing mode of aromatic amino acids and nucleic acids [18,30,31,32,33], ~1210—proteins, aromatic amino acids [34,35,36,37,38], ~1260—proteins, amide III [29,32,36,39], ~1666 cm^−1^—proteins, amide I α-helix [11,30,35,40,41].

Concerning serum samples, the univariate analysis reveals a broad area (350–1750 cm^−1^) in the spectra that can be used successfully in differentiating PCa from controls. This region is composed of the following 3 windows: 350–1550, 1600–1700, and 1750 cm^−1^. The univariate analysis indicates the presence of the following 16 important peaks for PCa and normal sample differentiation, located at: 390, 499, 643, 729, 815, 892, 1012, 1100, 1137, 1210, 1331, 1368, 1412, 1511, 1582, 1660 cm^−1^. These can be assigned as follows: 390—uric acid [29], 499—proteins, amino acids [29,40,42,43], 643—DNA bases, ring stretching of uric acid and hypoxanthine [18,31,36,37,43], 729—DNA/RNA bases, ring stretching in uric acid and hypoxanthine [30,31,43,44], 815—collagen, uric acid [18,41,45], 892—deoxyribose phosphate backbone, glutathione, uric acid [18,43,46], 1012—aromatic amino acids and nucleic acids [18,30,31,32,33], 1100—proteins, phospholipids and carbohydrates [36,38,43,46], 1137—aminoacids, phopsholipids [18,41,42,45], 1210—aromatic amino acids [34,35,36,37,38], 1331—nucleic acid bases, phospholipids, proteins, amide linkages [18,35,36], 1368—pyridine bases, amide III, phospholipids [36,47], 1412—collagen, lipids and phospholipids [31,36], 1511—DNA/RNA bases, amide II, phenylalanine [36], 1582—phenylalanine [18,31,36,43], DNA/RNA bases, 1660 cm^−1^—amide I α-helix [11,30,35,40,41]. Most of these bands are attributed to nucleic acid bases and proteins, which may indicate that a PCa complex metabolism has a crucial role in disease development. For a more accurate determination, we have prepared a tentative assignment in Appendix A for all the recorded SERS spectra of plasma and serum samples according to the available literature [19,29,30,31,32,33,34,35,36,37,38,39,40,41,42,43,44,45,46,47,48,49,50,51,52,53,54,55,56,57,58,59,60,61,62,63,64,65,66,67].

498, 642, 815, 893, 1010, 1137, 1210, 1368, and 1412 cm^−1^ vibrational bands show high intensities in PCa serum samples compared to normal samples. On the other hand, we notice that 729, 1100, 1331, 1511, 1582, and 1660 cm^−1^ show higher intensities in normal samples compared to PCa samples.

At first glance, we can observe that the following 4 major peaks are common in both serum and plasma samples: 390, 1010, 1210, and 1662 cm^−1^. The 1010 and 1210 cm^−1^ followed a similar increased pattern regarding intensities in PCa samples, while the 390 and 1662 cm^−1^ bands’ increased intensity is specific for normal samples. Moreover, an overview of both SERS spectra considering the results obtained from the univariate analysis indicates a very uniform tendency respecting the bands’ intensities.

In the case of PCa plasma and serum samples, 1010 cm^−1^ symmetric ring breathing mode of phenylalanine and 1210 cm^−1^ protein bands showed an increased signal than those of normal samples. On the other hand, the amide I bands (1600–1700 cm^−1^) were lower compared to normal plasma and serum samples. A slight difference can be observed in the case of amide III regions regarding the relevant peaks in discriminating between PCa patients and healthy donors. The 1260 cm^−1^ peak was determined to be relevant for normal plasma samples, where it presents a higher intensity than PCa samples. Concerning serum samples, the 1368 cm^−1^ peak showed an increased behavior for PCa compared to normal samples. Analyzing both SERS spectra, these two peaks followed similar intensities in both types of samples investigated.

These similarities between both plasma and serum samples may be due to abnormal metabolism associated with cancer, implying the activation of alternative metabolic systems to create ATP, proteins, nucleosides, and lipids for cellular growth [68]. Cancers of the peripheral prostatic epithelium may have a similar, citrate-oriented metabolism to that of normal prostate tissue. The oxidative phosphorylation in primary prostate tumors seems to be increased, although glycolysis is restricted. Prostate cancer also seems to be connected with the synthesis of fatty acids in the form of lipogenesis [69]. Advanced castrate-resistant prostate cancer is characterized by increased glycolysis. Maintaining the amino acid pool and converting it to glucose, lipids, and precursors of nitrogen-containing metabolites such as purines or pyrimidines for nucleic acid synthesis are all important aspects of amino acid metabolism in prostate cancer growth [70].

PCa has been previously linked to a buildup of cholesterol and has been shown to synthesize fatty acids by means of de novo lipid synthesis [71]. It is well known that freshly generated fatty acids enhance cellular pathways that promote cell growth and survival in cancer patients [72,73]. Lipogenesis has been demonstrated to increase the saturation of membrane lipids, which has implications for membrane dynamics and the absorption and effectiveness of chemotherapy [74].

The bands situated between 900 and 1300 cm^−1^ are mostly generated by carbohydrates and phosphates found in nucleic acids [36]. Carbohydrates are represented by the C-COO^−^ stretching vibration at the band 915 cm^−1^ [39,53]. Biopsies of prostate cancer tissue and cervical cancer tissue have shown that glycogen concentration is lower in prostate cancer tissue and cervical adenocarcinoma cells [75]. It is known that in malignant cells, the cause of decreased glycogen can be an implication of increased metabolic activity [75,76,77].

Due to the C=O and C-N stretching vibrations, the amide I band (1600–1700 cm^−1^) gives information on the secondary structure of proteins [78,79,80], which has been thoroughly studied in several research studies. There is, however, a correlation between the peak at 1617 cm^−1^ assigned to C=O, C=C, and NH_2_ stretching vibrations [30,42,59] and the existence of protein aggregates [79,81]. Protein misfolding and subsequent aggregation are caused by conditions that cause cancer cells to be subjected to stress [82]. Cancer cells and tumors may develop protein aggregates of the tumor suppressor gene p53 [83,84,85], which may play an essential role in the development of cancer [86]. It has been shown that the loss of proteostasis in cancer growth is linked to platinum resistance and the stem cell characteristics of certain ovarian cancers [87]. As an example, PNT1A and PNT2 normal prostate cell lines are characterized by spectral assignments of 1653 and 1636 cm^−1^, respectively, which are associated with α-helices and parallel sheets [71]. There are fewer β-sheets in proteins that are generally more soluble and less prone to congregating [88], which may be due to the fact that most proteins have a combination of β-sheets and α-helices as secondary structures [80].

## 5. Conclusions

The discrimination between PCa and healthy donors based on liquid biopsy still remains a challenging analysis since it needs multiple examinations. However, to our knowledge, this is the first study that has evaluated blood plasma and serum provided by PCa patients and healthy donors using a combined SERS, multivariate and univariate analyses in order to establish which analyte can offer a better diagnostic value. Our results show that serum samples have a better diagnostic capacity compared to plasma samples. The best values have been obtained when performing the multivariate analysis of the full spectrum as well as the two spectral intervals of 825–1050 and 1506–1750 cm^−1^.

The spectrum is dominated by aromatic amino acids (tryptophan, tyrosine, phenylalanine, serine) and protein vibrational bands, which have been shown to achieve a valuable potential as biomarkers [70,89]. There are several studies that engaged the use of chromatography and mass spectrometry techniques to determine the free amino acid profiles from liquid samples such as urine, serum, or plasma [89,90,91,92,93,94,95]. These are indicating that the amount of some specific amino acids in such biological samples may gain more insights into PCa diagnosis purposes. Therefore, we believe that a combination between SERS and proteomics and metabolomics methods together with multivariate and univariate analyses tools will elevate the standard diagnostic (PSA level evaluation and prostate tissue biopsy) properties.

There are some drawbacks to this research, the most significant of which being the limited number of patients included in the study. In order to obtain the most accurate results, we only included individuals with prostatic adenocarcinoma confirmed by biopsy. Our study also eliminated individuals with other chronic illnesses, patients with unconfirmed tumors, or patients with confirmed tumors but not eligible for surgery (advanced cancers), since their serology would have remained unchanged. However, since this is a pilot study whose major goal is to estimate average values and variability in order to design larger later investigations, we believe that the sample sizes were sufficient.

Our study shows promising results since SERS analysis can be performed on small amounts of liquid samples with high specificity and reproducibility as a direct consequence of the use of our solid plasmonic substrate [18]. Moreover, the implementation of multivariate and univariate analysis allowed us to determine that serum samples offer more accurate results in discriminating between PCa patients and healthy donors when compared to plasma samples. Another advantage of this design refers to the minimal invasiveness of the technique, which is supported by easy handling and fast results generation. At the same time, there is still a need for such investigations on large cohorts in order to establish the necessity of needle biopsy and histopathological examination.

## Figures and Tables

**Figure 1 cancers-14-03227-f001:**
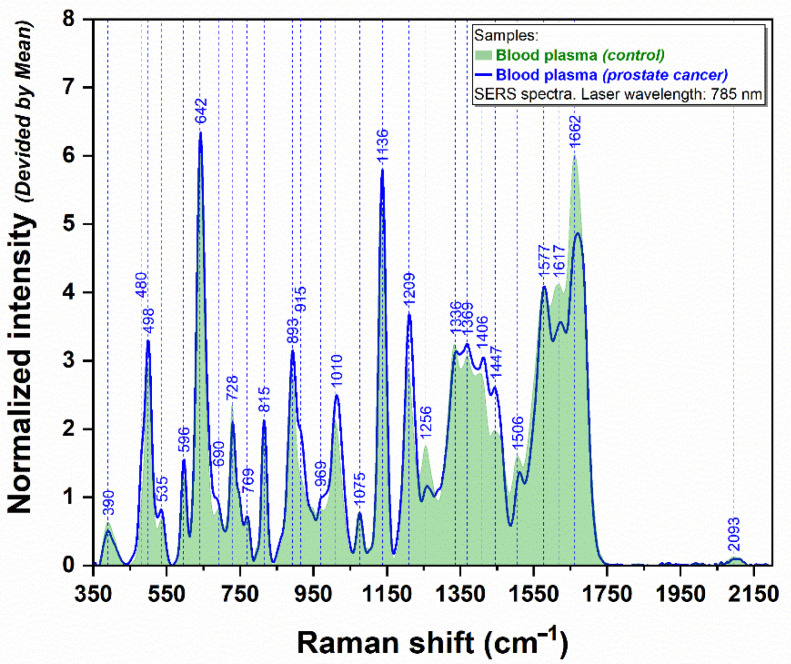
The average SERS plasmatic spectra obtained from PCa patients (*n* = 27, blue spectrum) and healthy donors (*n* = 14, green spectrum), using a 785 nm laser.

**Figure 2 cancers-14-03227-f002:**
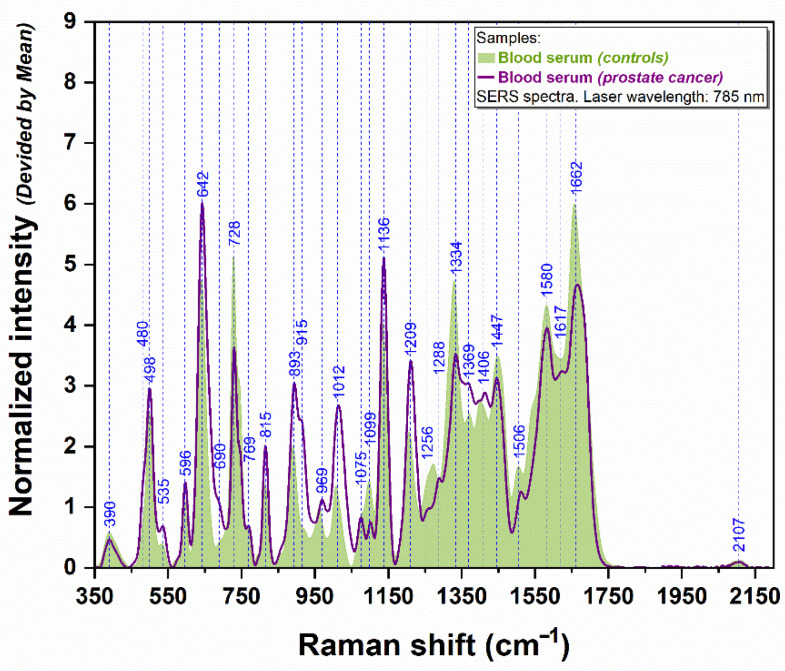
The average serum SERS spectra of PCa patients (*n* = 29, magenta spectrum) and healthy donors (*n* = 14, green spectrum), using a 785 nm laser.

**Figure 3 cancers-14-03227-f003:**
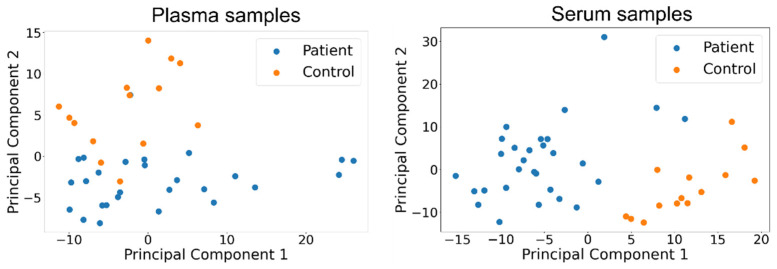
Score-score plots for the first two principal components obtained from plasma and serum samples LOOCV analysis.

**Figure 4 cancers-14-03227-f004:**
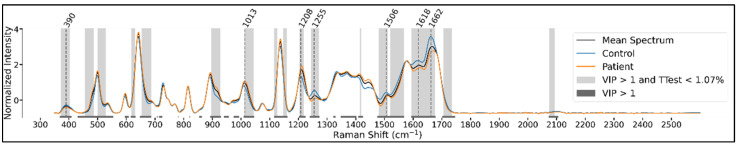
Mean spectra with emphasized t-test significance and VIP > 1 for plasma samples.

**Figure 5 cancers-14-03227-f005:**
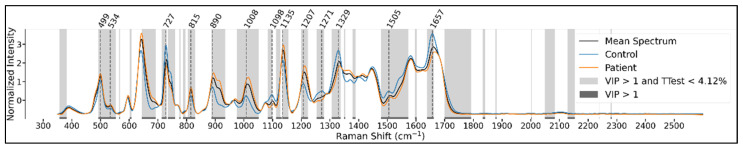
Mean spectra with emphasized t-test significance and VIP > 1 for serum samples.

**Table 1 cancers-14-03227-t001:** Clinical data of patients group.

Number of Patients: 29			
Age (years old)			
Min.	Max.		Mean
52	68		61
PSA (ng/mL)			
Min.	Max.		Mean
5.8	39.82		13.36
Pre-operative Gleason Score			
6		9 patients	
7(3 + 4)		12 patients	
7(4 + 3)		5 patients	
8		1 patient	
9		2 patients	
Post-operative Gleason Score			
N+		2 patients	
M+		0 patients	
L+		2 patients	
R+		4 patients	

Legend: N+ (node positive); M+ (positive metastases); L+ (lymphatic invasion); R+ (tumoral margins).

**Table 2 cancers-14-03227-t002:** PCA-LDA results on plasma and serum samples.

Sample	Accuracy	Precision	Sensitivity	Specificity	True Pos.	True Neg.	False Pos.	False Neg.
Plasma	87.8%	86.7%	96.3%	71.4%	26	10	4	1
Serum	97.7%	100.0%	96.6%	100.0%	28	14	0	1

**Table 3 cancers-14-03227-t003:** PCA-LDA results on plasma and serum samples for 1200–1700 cm^−1^ spectral region.

Sample	Accuracy	Precision	Sensitivity	Specificity	True Pos.	True Neg.	False Pos.	False Neg.
Plasma	80.5%	85.2%	85.2%	71.4%	23	10	4	4
Serum	93.0%	96.4%	93.1%	92.9%	27	13	1	2

**Table 4 cancers-14-03227-t004:** PLSDA results on plasma and serum samples.

Sample	Accuracy	Precision	Sensitivity	Specificity	True Pos.	True Neg.	False Pos.	False Neg.
Plasma	90.2%	89.7%	96.3%	78.6%	26	11	3	1
Serum	95.3%	100.0%	93.1%	100.0%	27	14	0	2

## Data Availability

The data presented in this study are available on request from the corresponding author. The processed data is contained within the article.

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
