# Peer review of "New Insights into the Multivariate Analysis of SER Spectra Collected on Blood Samples for Prostate Cancer Detection: Towards a Better Understanding of the Role Played by Different Biomolecules on Cancer Screening: A Preliminary Study"

_cancers, 2022, doi:10.3390/cancers14133227_

Round 1

Reviewer 1 Report

The author has made appropriate correction. No further request for this article.

Author Response

Dear Reviewer, 

Thank you for your time and effort in the revision process of our manuscript!

Regards

Prof. Rares Ionut STIUFIUC

Reviewer 2 Report

I am fine for this manuscript to be considered for publication.

Author Response

Dear Reviewer, 

Thank you for your time and effort!

Regards

Prof. Rares Ionut STIUFIUC

Reviewer 3 Report

1.The authors added PC spectra in the supplementary file. Although it’s not critical, it’ll be helpful to provide classification graphs (scatter plot with trained classifier), e.g. for classification using 2 PCs.

2.The authors added “The classification results of PCA-SVM using 2 PCs and a linear kernel for the SVM, SVM using a linear kernel and no dimensionality reduction and LDA with no dimensionality reduction are presented in table S3 and figures S1 and S2.”

-- According to the caption, Figure S1 includes first two principal components for plasma spectra, Figure S2 includes first two principal components for serum spectra. No classification is presented in Figures S1 and S2.

 “SVM using a linear kernel and no dimensionality reduction and LDA with no dimensionality reduction”

-- If no dimension reduction, does this mean the classification was performed using the intensity values of the entire spectra? Won’t this lead to overfitting? The results from LDA are all 100%, which may be due to overfitting.

3.       When PCA is used for classification, the leading components are not necessarily the best components for classification. It’s possible that other components carry better characteristic information and provide better classification. Please elaborate.

4.   There are a long history of cancer diagnosis using Raman by Dr. Robert Alfano. Some recent publications are shown below for your reference.

a. Binlin Wu, et al., "Statistical analysis and machine learning algorithms for optical biopsy", Proc. SPIE 10489, Optical Biopsy XVI: Toward Real-Time Spectroscopic Imaging and Diagnosis, 104890T (19 February 2018); https://doi.org/10.1117/12.2288089

b.        Yan Zhou, et al., "Invited Article: Molecular biomarkers characterization for human brain glioma grading using visible resonance Raman spectroscopy", APL Photonics 3, 120802 (2018) https://doi.org/10.1063/1.5036637

c.       Bendau, E., et al. Distinguishing metastatic triple-negative breast cancer from nonmetastatic breast cancer using second harmonic generation imaging and resonance Raman spectroscopy. J. Biophotonics. 2020; 13:e202000005. https://doi.org/10.1002/jbio.202000005

5.   AUROC should be “Area Under the Receiver Operating Characteristic” curve

Ref: Jain M, et al., Exploring Multiphoton Microscopy as a Novel Tool to Differentiate Chromophobe Renal Cell Carcinoma From Oncocytoma in Fixed Tissue Sections. Arch Pathol Lab Med. 2018 Mar;142(3):383-390. doi: 10.5858/arpa.2017-0056-OA. Epub 2017 Dec 8. PMID: 29219617.

Author Response

Dear Reviewer, 

please find as attached files a modified version of the manuscript together with our point by point responses to your valuable comments. 

Thank you again for your time and effort.

Regards 

Prof. Rares Ionut STIUFIUC 

Reviewer 4 Report

The authors responded to the reviewer's comments and made changes to the text of the manuscript. However, given the small sample size (29 patients with prostate cancer and 14 healthy controls), the results should still be considered preliminary. I would encourage authors to include this information in the title.

Author Response

Dear Reviewer, 

please find as attached file our response to your comments.

Thank you for your time and effort!

Kind Regards

Prof. Rares Ionut STIUFIUC

This manuscript is a resubmission of an earlier submission. The following is a list of the peer review reports and author responses from that submission.

Round 1

Reviewer 1 Report

The author presented the new diagnostic system using SER spectra collected on blood samples for Pca. 

Major

Although technology may be new, with high specificity and high specificity, the current module has just had prerimilary data, taht may not be sufficient to be published as an original article. 

Further updates will be ideal. 

Author Response

Dear Reviewer, 

thank you for the time and effort you dedicated for the improvement of our manuscript.  

Please find as attached file the response to your observations. We have also uploaded a revised version of the manuscript/SI file. The modifications are highlighted in yellow. 

Regards

Rares Stiufiuc 

Reviewer 2 Report

The authors demonstrated the use of Raman and its counterpart Surface Enhanced Raman Spectroscopy (SERS) can be used on plasma or serum from prostate cancer patients to predict disease occurrence. The method used involved the addition of nanoparticles to plasma or serum to enhance Ramen signals. They seem to have an extraordinary sensitivity and specificity with the SERS technology. The multivariate analysis was performed.  There are a few issues that need to be addressed:

1) It is not clear what the values were that were used to perform the statistical analysis for the various spectral profiles.

2) It would be good to show a comparison in a ROC type format a comparison of the spectral values with the PSA. In addition, it would be good to the ROC when PSA is combined with with the the Ramen spectral values.

3) It would also be nice to see the time dependance of the predictability of the SERS. Is there a significant time difference in the SERS-based detection and clinical diagnosis for prostate cancer.

Author Response

(The authors gave the same response as above.)

Reviewer 3 Report

1. I did not find information about healthy donors, for example, age. Were there significant differences in age between the groups of donors and patients with prostate cancer? Could the donors have had signs of dysplasia (PIN I or PIN II)? Since there is no data on the control group, it is impossible to assess the correctness of the data presented in the article. 2. Have you studied the correlation of spectra with the level of prostate-specific antigen (PCA) in the group of patients with prostate cancer? Since the authors use serum and plasma, such a correlation may occur. 3. Do the characteristics of the spectra change depending on the severity of the disease?

Author Response

(The authors gave the same response as above.)

Reviewer 4 Report

The manuscript “New insights into the Multivariate Analysis of SER spectra collected on blood samples for Prostate Cancer Detection: towards a better understanding of the role played by different biomolecules on cancer screening” by Munteanu et al. evaluated SERS Raman spectra of blood plasma and serum from PCa patients and healthy donors using multivariate and univariate analyses methods and machine learning. The authors compared different multivariate methods, classification methods, and Raman bands, and the spectra from plasma and serum, to find possible biomarkers for PCa liquid biopsy. Although the method is not novel, the results are promising. The following questions should be addressed before the manuscript can be considered for publication.

Provide more details about the blood sample collection process. How are the patients and healthy people prepared? Do other physiological conditions and even diet affect the blood samples?

I suggest the authors provide more results such as graphs of PC spectra, and classification using SVM and LDA.

Line 178: “For SERS measurements 1 μl of serum, respectively 1 μl plasma, were poured on the top of plasmonic substrates and were left to dry for 30 minutes at room temperature before acquiring the SERS signal.” – what if measuring the liquid directly?

Line 180: “Both spectrum types were recorded at maximum 50 μm distance from the sample ring edges.” – what does this mean? Could you explain what “analyte ring edges” are?

Line 186: “exposure time 5 s and 4 accumulations”, “The final spectrum represents the average of 20 spectral acquisitions.” – Is each spectrum an average of 4 accumulations from the data acquisition? Did you take 20 spectra from each sample and then take the average as the final spectrum?

How many samples were collected from each patient? How many spectra were collected from each sample? How many spectra in total did you collect for analysis?

Line 200: “As a preprocessing step for the multivariate analysis, we align the spectra by sampling at equal 1 cm-1 intervals” – Were the raw spectral data collected at different wavenumber values?

Line 210: “optimizing for the best ratio of between-class and within-class variances” – Does "best" mean "maximum"? Then maybe it’s better to say maximum for clarity.

Line 238: “normalized to the integrated area under curve”; line 201: “normalizing them using the Standard Normal Variate method, where each spectrum's intensities are scaled and offset such that they have zero mean and unit standard deviation.” – Were two different normalization methods used?

“serum samples offer a better separation between controls and PCa groups”, “the use of serum samples instead of plasma ones can improve the discrimination” – Any explanation to the reason why?

Add a reference for “Benjamini-Hochberg (BH)-corrected T-Test”.

Why only two PCs were used? What if more PCs are used?

For cancer diagnosis using Raman or general optical spectroscopy, spectral analysis using PCA/PLS and SVM/LDA methods, and even liquid biopsy using blood samples for cancer diagnosis, the authors should cite works by Alfano group.

We have performed a univariate analysis, where we test the separability hypothesis at each sampled wavenumber using a T-Test.” – It’s very common to use peak ratios of key biomolecules. Refer to papers by Alfano group.

Spell out and describe AUROC. Describe how you calculated AUROC using the 6 major regions.

Spell out the abbreviations such as PSA, PCa, PCA-LDA, when first used.

Typos and grammatical errors such as: 

Line 33, "assist prevent” may be changed to “assist in preventing”; line 93, “which role” should be changed to “whose role”; line 106, “capable to generate” should be changed to “capable of generating”; line 126, change “recorder” to “recorded”; line 133, change “By the end” to “In the end”; line 255, “slightly differences” should be slight differences; line 467, correct “aminocids”.

Data Availability Statement: We choose to exclude this statement, the study did not report any data.” – I wonder if this is an appropriate statement here. Even though no raw data are included, are the graphs and/or results considered data?

Author Response

(The authors gave the same response as above.)
